



# Climate records in ancient Chinese diaries and their application in historical climate reconstruction—A case study of *Yunshan Diary*

Siying Chen[1,2], Yun Su[1,2], Xiuqi Fang[1,2], Jia He[1,2]

[1]Faculty of Geographical Science, Beijing Normal University, Beijing 100875, China

[2]Key Laboratory of Environmental Change and Natural Disaster, Ministry of Education, Beijing Normal University, Beijing 100875, China

*Correspondence to*: Yun Su (suyun@bnu.edu.cn)

**Abstract.** Private diaries are important sources of historical data for research on climate change. Their advantages include a high veracity and reliability, accurate time and location information, a high temporal resolution, seasonal integrity, and rich

content. In particular, these data are suitable for reconstructing short-term, high-resolution climate series and extreme climatic events. Through a case study of *Yunshan Diary*, authored by Bi Guo of the Yuan Dynasty of China, this article demonstrates how to delve into climate information in diaries, including species distribution records, phenological records, daily weather records, and perception records. In addition, this article considers how to use these records, supplemented by other data, to reconstruct climate change and extreme climatic events on various time scales, from multi-decadal to annual to

daily. The study of *Yunshan Diary* finds that there was a relatively low amount of precipitation in central and southern Jiangsu Province in the summer of 1309; the winter of 1308–1309 was abnormally cold in the Taihu Lake Basin. In the early 14[th] century at the latest, the climate in eastern China had begun to turn cold, which reflects the transition from the Medieval Warm Period to the Little Ice Age.

## 1 Introduction

Reconstruction of past climates can help better understand and respond to current climate change and predict future scenarios (PAGES, 2009). Historical documents are one of the primary sources of proxy data used to reconstruct past climatic information. Compared to natural proxy data (e.g., tree rings, ice cores, and stalagmites), historical documents are advantageous in their spatial coverage, temporal resolution, dating accuracy, location accuracy, and clarity of climatic significance (Zheng et al., 2014).Globally, most such historical documents are in China, Japan, and Europe (Pfister et al.,

2008). Thanks to its long history, China has enormous quantities of documentary data, including abundant climate-related records which are uninterrupted for the past two millennia. This is China's exceptional advantage in historical climate research (Man, 2000).

There are four types of Chinese historical documents that contain climate information. These are (1) the traditional documents classified as *jing* (classics), *shi* (histories), *zi* (philosophies), and *ji* (anthologies); (2) local gazettes compiled by





magistrates; (3) archives of the Ming and Qing Dynasties, represented by *Qing Yu Lu* (The Records of Sunny or Rainy Days) and *Yu Xue Fen Cun* (The Records on Rainfall Infiltration and Snowfall); and (4) private notes and diaries (Zhang, 1996; Ge et al., 2018). In China, diaries can be traced back two millennia to the Western Han dynasty. There are more than 1,000 surviving ancient diaries (Chen, 2004), of which approximately 200 contain weather or weather-related records (Ge et al., 2018). Currently, private diaries of the Ming and Qing Dynasties, mostly authored by officials or men of letters, are used as

one of the primary sources in historical climate research. However, compared to studies based on other documentary data (e.g., history books, local gazettes, and archives), historical climate studies based on data in diaries are relatively limited and mostly focussed on the Little Ice Age (LIA). Studies have been done mainly based on weather, phenological, and perception records in diaries. The available results involve a number of areas, including the reconstruction of temperature series, extreme cold events, and the characteristics of temperature change(Chu, 1973; Fang et al., 2005; Liu and Man, 2012; Xiao

et al., 2006; Zheng et al., 2015), the reconstruction of precipitation and the East Asian rainy season (also known as the *Meiyu* season) (Man et al., 2007; Xiao et al., 2008; Yan et al., 2011; Zhang et al., 2011b, 2013a), dust weather (Fei et al., 2004, 2005, 2009; Zhang et al., 2006; Yang et al., 2013), and adaptive human behaviours (Zhang et al., 2007b). Researchers have also discussed the veracity and reliability of weather and climatic records in diaries (Zhang et al., 2007a, 2013a; Fei et al., 2009).

In Europe, weather diaries have been used to study temperature (Nordli, 2001), precipitation (Gimmi et al., 2007; Pfister et al., 1999), droughts (Linderholm and Molin, 2005), and climatic effects of volcanoes (Lee and Mackenzie, 2010) in historical periods, particularly the period from the 16[th] to the 19[th] century. These diaries were mostly written by priests, farmers, and scholars. For example, based on the weather records in the diary of an Italian priest, Raicich (2008) analysed temperature and precipitation conditions in Trieste in the period 1732–1749. Based on an English farmer's diary, Lee and

Mackenzie (2010) identified climate abnormities in England in the two years following the eruption of Mount Tambora in 1815. In India, Adamson and Nash (2013, 2014) studied the onset date of the summer monsoon as well as monsoon precipitation based on private diaries and other documents. In Japan, diaries have been used to study summer temperatures (Mikami, 2008), the winter monsoon (Hirano and Mikami, 2008), and typhoon weather (Grossman and Zaiki, 2009). Diaries have also been used to investigate historical climate in other regions, including Australia (Gergis et al., 2012) and Africa

(Grab and Nash, 2009; Nash and Grab, 2010).

Through a case study of a 14[th]-century diary, *Yunshan Diary*, this article illustrates the types and characteristics of historical climatic information recorded in ancient Chinese diaries and demonstrates how to use these records to analyse the characteristics of weather and climatic events as well as climate change on various time scales.





## 2 Data sources

**2.1 Brief introduction to *Yunshan Diary***

The raw materials used in this study originated from *Yunshan Diary*, which is included in *A Series of Diaries of the Jin and Yuan Dynasties* published by Shanghai Bookstore Publishing House (Gu and Li, 2013). *Yunshan Diary* was written by Bi Guo (1280–1335), also called Tianxi Guo, a calligrapher and painter of the Yuan Dynasty. Guo resided in his hometown, which is present-day Zhenjiang in Jiangsu Province, China (Yu, 1989). During the period recorded in his diary, Guo

travelled multiple times on official business or to visit friends. The scope of travelling involves central and southern Jiangsu Province and northern Zhejiang Province, primarily within the Taihu Lake Basin (TLB) (Fig. 1).

Situated in the Yangtze River Delta along the south-eastern coast of China, the TLB (119°3′-121°54′E, 30°7′-32°14′N) encompasses an area of $3.6 \times 10^4$ km$^2$. Except for a few low mountains and hills in the western region, it is mostly dominated by flat plains (altitudes below 10 m). With the highest drainage density in China, the TLB is home to numerous rivers and

lakes, and the Taihu Lake is China's third largest freshwater lake. The TLB has a subtropical monsoon climate. North winds prevail in the cold and dry winter, while southeast winds prevail in the hot and rainy summer. The annual average precipitation here is 1,000–1,400 mm, the annual average temperature is 15–16 °C, and the average temperature in January is 1.5–4 °C (Huang, 2000; Wu et al., 1993; Xu et al., 2017).

Covering a total of 16 months, from 12 September 1308 to 2 December 1309, *Yunshan Diary* contains daily records of Guo's

work and life, together with weather records on most days. In this study, the dates in the diary (originally on the lunar calendar) were converted to dates on the Gregorian calendar according to *The Chinese Almanac for Two Thousand Years* (Perpetual calendar editing group, 1994). All the dates in this article are Gregorian dates. The ancient place names in the diary were converted to the corresponding present-day city or county names according to *The Historical Atlas of China* (Tan, 1982).

**2.2 Other data**

For quantification and comparison purposes, modern meteorological data from instrumental measurements were used in this study. These data originated primarily from the collections of early instrumental data (The reference room of Beijing meteorological center, 1984) and the China Meteorological Data Service Center (http://data.cma.cn/). All the data were published by the National Meteorological Information Center of China Meteorological Administration.

Modern phenological data originated primarily from the collections of observation data of the Chinese Phenological Observation Network (Wan, 1986; Wan and Liu, 1986). Some phenological data about meteorological events were extracted from the Daily Surface Climate Dataset for China (V3.0).



## 3 Extraction of historical climatic information and the reconstruction of weather and climate

Based on the content, climate records in historical documents can be classified into four categories, namely, weather records,
meteorological disaster records, phenological records, and records relating to the characteristics of regional climate (Zheng et
al., 2014). All the records relating to weather and climate in *Yunshan Diary* were excerpted based on a comprehensive read.
In addition, the time and location of each weather or climatic event were recorded.

Four types of information were extracted from *Yunshan Diary*, namely, species distribution records (one), phenological
records (six), weather records (342), and perception records (87).

### 3.1 Species distribution records and their climatic significance

### 3.1.1 Species distribution records

The distribution range of plant and animal species is adapted to the zonal climatic characteristics of the region and reflects
the average climatic conditions over decades (Zheng et al., 2014). The species distribution range is limited by environmental
conditions (Gong et al., 1983a). If there is a definite and predominant climate-limiting factor for the survival of a certain
species, its distribution range may be an indicator of the climate (Man, 2009). For example, the planting areas of zonal crops
and fruit trees may indicate the climatic characteristics of the region on a multi-decadal scale. Phased climate change and the
movement of the climatic zone can be deduced by comparing the planting areas between time periods (Zheng et al., 2014).

The study area is located in the lower reaches of the Yangtze River and belongs to the North Subtropical Zone. Central and
southern Jiangsu Province is close to the northern boundary of the North Subtropical Zone, one of the regions that is most
sensitive to climate change in China (Zhang, 1996). Within this region, the main species with notable climatic significance
are subtropical crops, animals, and plants. (1) The northern planting boundary of double-cropping rice roughly corresponds
to ≥10 °C active accumulated temperature of 4,800 °C (Zheng et al., 2014). The lower reaches of the Yangtze River are close
to the northern planting boundary of double-cropping rice. Since the Tang dynasty, the rise and fall of double-cropping rice
in the study area has approximately corresponded to climate change in terms of temperature (Man, 2009; Zhang, 1996). (2)
For subtropical plants, such as citrus and tea trees, their growing areas are primarily limited by winter temperatures and
freezing injury. The northern planting boundary of these plants is close to the northern boundary of the Subtropical Zone
(Man, 1999; Man and Yang, 2014; Zhang et al., 2019). (3) For subtropical animals, such as rhizomys, elephants, buffalos,
and Chinese alligators, their distribution ranges are affected by human activities to a large extent. As a result, the distribution
range of animals is less accurate than that of plants as climate indicators (Gong et al., 1983a).

One species distribution record was excerpted from *Yunshan Diary*, which was recorded during Guo's sojourn in Hangzhou
in 1308. It reads "there are several dwarf citrus trees outside the window, bearing countless fruits, which weighed down and
almost broke the branches; there are no such trees in my hometown".



### 3.1.2 Climatic significance of the species distribution record

Citrus is a typical kind of subtropical perennial fruit, which prefers a warm, humid climate. It is sensitive to low
temperatures, and can be easily hit by freezing injury. The winter minimum temperature is the primary factor that limits the
northern planting boundary of citrus (Man, 1999), which cannot go past the isoline of a multi-year average extreme
minimum temperature lower than -9 °C (Zheng et al., 2014). Citrus is native to China. China has a history of nearly three
millennia of citrus tree cultivation. Thus, there are rich, comparable records for citrus planting locations in the historical
documents. Therefore, in China, the northern planting boundary and southern freeze-to-death boundary of citrus are widely
used indices in the study of climate change on a long time scale (Chu, 1973; Gong and Zhang, 1983; Man, 1998; Zhang et al.,
1977; Zhang, 1996).

Man systematically collated the changes in the northern planting boundary of citrus between the Spring and Autumn Period
and the Qing Dynasty and discussed their correlation with climate change (Man, 1999). During the Medieval Warm Period
(MWP), the citrus planting area reached at least Nanyang (33.0°N) in Henan and Nanjing (32.2°N) in Jiangsu in the mid-13th
century. Orange trees, which are relatively cold-resistant, were once planted in Jiaozuo (35.2°N) in north-western Henan. In
the mid-Ming Dynasty, citrus planting records only appeared in Shanghai (31.3°N) and Taicang (31.5°N) in the Yangtze
River Delta. Further north, there were only records of orange plantings in Dantu (32.1°N), Tongzhou (32.0°N), and Rugao
(32.3°N). In the early Qing Dynasty, no citrus planting records are available in Shanghai.

According to the records in *Yunshan Diary*, in the early 14th century, citrus was planted in Hangzhou (30.2° N), whereas no
citrus was planted in Zhenjiang (32.2° N), Guo's hometown. The latitude of Zhenjiang is similar to that of Nanjing.
Evidently, the northern planting boundary of citrus was more southern in this period than in the MWP. This suggests that the
climate had begun turning from warm in the MWP to cold.

### 3.2 Phenological records and their climatic significance

### 3.2.1 Phenological records

Phenology refers to the relationships between periodical biological phenomena and environmental conditions (Chu and Wan,
1963). Phenological phenomena are natural phenomena that are affected by climate, hydrology, and soil and occur quasi-
periodically with a repetition period of one year. There are three main types of phenological phenomena, namely,
phenological phenomena of plants (including the sprouting, leafing, flowering, and abscission of woody plants and the
sowing, farming, and harvesting of crops), phenological phenomena of animals (including the arrival, first warble, last
warble, departure, and hibernation of migratory birds, insects, and other animals), and the periodicity of meteorological or
hydrological events (including the first frost, last frost, first snow, last snow, and freezes and thaws of rivers and lakes).

Advances or delays in phenological phases are primarily affected by climatic factors, particularly temperature. The basic
approach for reconstructing historical climate change based on phenology is as follows. The phenological dates in historical





records are compared with those of the same phenological phenomena at the same location in modern times, and the
differences of the climate factors between the ancient and present times are deduced from the relationship between
phenology and climate (Liu et al., 2017). Three factors, namely, time, location, and phenological event, need to be clarified
when constructing historical climate based on phenological records. Compared to records from other data sources (e.g.,
agricultural books and poems), it is relatively easy to verify the time and location of records in private diaries, but the
phenological events require careful textual research.

In this study, potential phenology-related records were identified in *Yunshan Diary* and further examined to determine their
veracity. For example, on 6 October 1308, Guo was in Hangzhou and wrote the following in *Yunshan Diary*: "There was a
frosted moon all over the sky and chilly air." While he mentioned "frosted" and "chilly", this record is insufficient to suggest
that the first frost in Hangzhou in 1308 occurred on 6 October. First, "frosted moon *(霜月)*" is a fixed phrase in Chinese,
which appears more than 40 times in *Complete Tang Poems* alone. "Frosted moon" refers to a cold night's moon. In this
term, the moon is likened to frost to highlight its pure white colour and coldness (Chen, 2018). For example, in one of his
poems, Zhenbai Wang of the Tang dynasty wrote "a frosted moon is setting".  Second, no "frost"-related records appear
again in the diary over the period of more than a month between 6 October, when Guo wrote "a frosted moon all over the
sky" in his diary, and 11 November, when Guo left Hangzhou. Instead, there are some records relating to hot weather in the
diary. For example, Guo wrote, "The inn was very warm, and my clothes were drenched in sweat, so I kept shaking a fan,"
on 10 October and, "It rained again, humid and hot," on 25 October in his diary. Therefore, the "frosted moon" in the
abovementioned record is a literary expression and should not be used as evidence for the phenological event of first frost.

A total of six phenological records were identified from *Yunshan Diary* (Table 1).

The species corresponding to the phenological records of animals and plants in *Yunshan Diary* were identified, and their
present-day names were determined. In addition, the phenological phases were defined according to the text description in
the diary on the basis of modern phenological observation methods. Here, the sixth phenological record in Table 1 is used as
an example. In *Yunshan Diary*, there is a record of the blooming of peach and plum flowers in Zhenjiang on 29 March 1309.
The relevant record reads, "At dusk, I went out of the gate of the Ganlu Temple. I looked outside of the Dingbo Gate, seeing
peach and plum flowers, red and white." In China, *Prunus davidiana French* is distributed primarily in the middle and lower
reaches of the Yellow River, whereas *Prunus persica (L.) Batsch* is distributed primarily in the Yangtze River Basin and
Huaihe River Basin (Gong et al., 1983a). The modern full flowering stage of *Prunus persica (L.) Batsch* occurs between 14
March and 15 April(Wan, 1986). Based on the geographical distribution and phenological phase, it can be deduced that the
"peach" in the diary refers to *Prunus persica (L.) Batsch*. The modern unified observed flowering stage of woody plants is
divided into three sub-stages, the first flowering stage, the full flowering stage, and the end of flowering stage. The full
flowering stage for woody plants is defined as "the stage when petals have unfolded from more than half of the flower buds
on the observed trees" (Wan and Liu, 1979). According to the record in *Yunshan Diary*, Guo stood by the gate of the Ganlu
Temple and looked outside of the Dingbo Gate. The Ganlu Temple is approximately 1 km away from the historic site of the



Dingbo Gate. The fact that Guo could see a scene of "peach and plum flowers, red and white" from such a distance suggests that the flowers must have been in high bloom and met the aforementioned requirement—"petals have unfolded from more than half of the flower buds". Therefore, it can be deduced that *Prunus persica (L.) Batsch* was in the full flowering stage on 29 March of that year.

### 3.2.2 Climatic significance of the phenological records

Of the six phenological records in *Yunshan Diary*, one was from the fall of 1308, two were from the winter of 1308, two were from the spring of 1309, and one was from the summer of 1309. Corresponding phenological phases in modern times at the same locations can be found for three of the records. A comparison can help deduce the climatic conditions at that time.

The first snow in Huzhou in the winter of 1308 occurred on 14 December, 17 days earlier than the modern (1957–1979) average first-snow date of 31 December. According to the climate formation mechanism, an advance in the first-snow date in China is often related to the time of the southward movement of cold air masses from their source region, Siberia (Zheng et al., 2005). This suggests that the winter of 1308–1309 was relatively colder than usual, with stronger winter monsoon winds and a lower average temperature.

The last snow in Zhenjiang in the spring of 1309 occurred on 9 March, compared to the present-day average last-snow date of 7 March (Wan, 1986). The full flowering stage of *Prunus persica (L.) Batsch* in Zhenjiang in 1309 occurred on 29 March, compared to the present-day average full flowering date of 1 April (Wan, 1986; Wan and Liu, 1986). These two phenological phenomena can both reflect the temperature in spring. The last-snow date is primarily related to the time of reductions in the activity of cold air masses (Zheng et al., 2005). The time of the phenological phases of plants in spring is closely related to the temperature in the period before. The higher the temperature is, the more rapidly plants develop, and the earlier the phenological phenomenon occurs (Gong et al., 1983b). There are no significant differences in the last-snow date and full flowering date of *Prunus persica (L.) Batsch* between the spring of 1309 and the present time. This suggests that the temperature in the spring of 1309 was close to the modern average spring temperature.

### 3.3 Daily weather records and perception records and their climatic significance

### 3.3.1 Weather records

In addition to recording his daily activities, Guo often briefly described the weather conditions of the day. Generally, weather conditions were recorded on a daily time scale and described using simple words, like cloudy, sunny, rainy, and snowy. Changes in weather conditions within a day were sometimes recorded, and precipitation was also sometimes described in detail. Weather records were excerpted on a daily time scale. They can be classified into several types, including sunny, cloudy, precipitation (rainy or snowy), other, and missing (no records about weather). When no direct weather records are available, deductions can sometimes be made based on other text descriptions. For example, a sunny day can be deduced from a description of Guo's outdoor activities in the open air.



There are a total of 342 weather records in *Yunshan Diary*, and the integrity rate is 67.8%. Table 2 summarizes the monthly statistics of each weather type in *Yunshan Diary*.

### 3.3.2 Perception records

In addition to recording objective weather phenomena, Guo sometimes recorded his subjective perceptions of the weather conditions. These records are scattered and appeared primarily on days with relatively extreme weather conditions or a sudden change in weather. A total of 87 perception records were extracted from *Yunshan Diary*. These records can be classified into three types, namely, perceptions of temperature changes (mainly represented by "cold", "hot", "warm", and "cool"), the effects of weather conditions on humans' lives (e.g., "clothes drenched in sweat" and "unable to sleep in the cold night"), and humans' responses to weather changes (e.g., "it has been very cool for several days, so I put on clothes with lining"). It is difficult to quantify and compare perception records. Thus, they are only used as qualitative indices to supplement weather records.

### 3.3.3 Characteristic analysis of monthly or seasonal precipitation based on daily weather records

By continuous tracing daily weather records, it is possible to reconstruct a certain weather event or weather conditions in a certain month or season.

In eastern China, there is a positive correlation between monthly precipitation and monthly number of precipitation days. The number of precipitation days for each month can be determined by analysing the diary, so it is possible to estimate the precipitation based on the correlation. Here, the Gregorian months when Guo stayed more than 20 days in Zhenjiang in 1309 are selected for analysis. Let P= monthly precipitation. It is worth noting that Guo frequently took excursions and rarely stayed an entire month in Zhenjiang and thus missed some precipitation events. For comparison purposes, assuming that precipitation days are evenly distributed within a month, the proportion of precipitation days (R) is defined as follows. For the instrument-measurement period, R=the number of precipitation days in Zhenjiang/total number of days of the month. For 1309, R=the number of precipitation days in Zhenjiang recorded in *Yunshan Diary*/total number of days when Guo stayed in Zhenjiang in that month.

Of the modern meteorological stations, the Gaoyou station, which is the station closest to Zhenjiang, was chosen for analysis. The instrument-measured data obtained at the Gaoyou station for the period 1955–2018 demonstrate a significant positive correlation between P and R in each month, with a confidence level of 99.9%. The correlation coefficients range from 0.55 to 0.9. Table 3 summarizes the linear regression equations established between P and R for each month. When the independent variable (R) is zero, the dependent variable (P) should also be zero. Therefore, the linear regression model with no intercept is selected. Figure 2 shows the reconstructed results for monthly precipitation.

Here, the precipitation characteristics in the summer of 1309 are described briefly. In June 1309, there were seven precipitation days in Zhenjiang, which is 4.5 days less than the modern average. The reconstructed precipitation in Zhenjiang




for June 1309 is 102.3 mm, which is 40% lower than the modern average. In July 1309, Guo primarily stayed in Xinghua.
The integrity rate of weather records for this month is low. Nevertheless, terms such as "extremely hot" and "unbearably hot" can be found on multiple occasions in the diary for this month. This suggests that the temperatures were relatively high throughout this month. In addition, only two precipitation days were recorded in the diary. On days without direct weather records, Guo often did outdoor activities, such as excursions, boating, and resting in the shade (e.g., "Mr. Zhan and I rowed a boat into the lake to enjoy the cool but were frustrated by the mosquitos" on 25 July). Therefore, it can be deduced that most
of the days without weather records were not precipitation days. Clearly, there were high temperatures and a low precipitation in Xinghua in July. In August 1309, Guo primarily stayed in Zhenjiang. Few days with precipitation records can be found in the diary for this month. The first precipitation record—"It rained so I felt happy"—is found in the entry for 25 August. This is likely because there had been no rain in the nearly 20 days before that. In summary, there was a relatively low precipitation in central and southern Jiangsu in the summer of 1309.

It is necessary to note that the precipitation events recorded in Guo's diary were those that he observed. Some precipitation events might have been too slight for humans to notice. Gimmi et al. (2007) believed that the lowest daily precipitation that humans can perceive is 0.3 mm. Therefore, the number of precipitation days extracted from the diary may be less than the actual number. In addition, the integrity rate of weather records is quite low in some months, which may increase the uncertainty of the conclusion.

**3.3.4 Reconstruction of cold wave processes in winter based on daily weather records**

There are a number of records, often vivid and detailed, relating to cold-weather phenomena, such as low temperatures, rain, snow, and freezes, in *Yunshan Diary* for the period December 1308–February 1309. These records are highly valuable for understanding the climatic characteristics of the winter of 1308–1309. In this winter, Guo stayed in the TLB. Specifically, he stayed in Huzhou from 1 December to 30 December and in Zhenjiang and Changzhou from 31 December to 28 February.
Zhenjiang and Changzhou are geographically close and have similar wintertime climatic characteristics. Thus, they are viewed as the same region for analysis in the following study. Table 4 summarizes the daily weather conditions derived from *Yunshan Diary* in the winter of 1308-1309.

A cold wave is a weather event leading to a dramatic decrease in temperature, which is caused by the large-scale invasion of cold air from high latitudes to middle and low latitudes. According to the modern national standard in China, *Cold Wave*
*Levels* (GB/T21987-2017), a cold wave refers to a cold-air event that results in a decrease in the minimum temperature in a location by ≥8 °C within 24 h, ≥10 °C within 48 h, or ≥12 °C within 72 h and causes the minimum temperature in the location to be ≤4 °C. The main weather characteristics of the invasion of a cold wave into Jiangsu include temperature decreases, strong winds, rain, and snow. The invasion of a cold wave also results in a probability of precipitation of 92% (The editorial board of "The climate of Jiangsu Province" from Jiangsu meteorological bureau, 1992). Cold waves in late fall





or early spring often cause frosts, while cold waves in the dead of winter may lead to freezing rain and the freezing of rivers, lakes, and ports (Zhang et al., 2011a). There will be a rise in temperature after a cold wave.

It is impossible to deduce the extent of decrease in temperature based on the textual records in *Yunshan Diary*. Nevertheless, multiple records for weather phenomena (e.g., rain, snow, frosts, freezes, and strong winds) closely related to cold waves can be found in the diary. Thus, periods in which these representative weather phenomena continuously occurred were extracted

from the diary. On this basis, together with Guo's perceptions of temperature changes, it is deduced that at least four notable cold waves occurred in the TLB in the winter of 1308–1309 (highlighted in blue in Table 4). The third and fourth cold waves were extremely strong. In addition, there was a notable increase in temperature between two contiguous cold wave events, predominantly reflected by consecutive sunny days and subjective perceptions such as "warm".

The third cold wave occurred between 1 and 5 January 1309. During this time, Guo was travelling on a boat back to

Zhenjiang from Wuxi via the Jiangnan Canal, a section of the Beijing-Hangzhou Canal. On 1 January, "A northeast wind broke out, and it was extremely cold"; and it began to snow at night. On 2 January, the weather remained "bitterly cold". In addition, the oar began to ice over, affecting the sailing. On 3 January, "Ice had closed in from all directions", and the canal was completely frozen. This rendered it impossible to sail the boat. The boatmen broke the ice and moved it onto the canal bank. Eventually, "piles of ice, two or three *chi* tall, were accumulated on the canal bank". Evidently, there was a

considerably thick ice layer in the canal. On 4 January, the weather cleared up, but the temperature remained very low. "There was thick ice in the canal, and we could not move forward." On 5 January, it remained impossible to sail the boat. Guo had to abandon the boat and ride a horse home. Manifestly, the temperature was extremely low during this cold wave, and the Jiangnan Canal was frozen for at least three days.

The fourth cold wave occurred between 1 and 10 February 1309. There were eight consecutive days of snowfall from 1 to 8

February. The snowfall was so heavy that Guo could not leave the house. In addition, there was a notable phenomenon of snow cover. There is a snowfall, snow cover, or icing record for every day between 3 February, when "Snow-covered trees connected with one another", and 12 February, when "The snow melted." Thus, it can be deduced that snow cover lasted for approximately 10 days. The snow cover was so deep that "No roads could be seen in all directions."

## 4 Discussion

### 4.1 Climate change indicated in *Yunshan Diary*

The following climatic characteristics can be derived from the above analysis. The climate in eastern China was colder in the early 14th century than in the mid-13th century. In the summer of 1309, there was relatively low precipitation in southern Jiangsu. In the winter of 1308–1309, it was abnormally cold in the TLB. On this basis, a combination of the records in the diary and other historical data or modern instrumental-measured data can help deduce the actual climate conditions at that

time. A combination of multiple cases can help improve climate reconstruction for a certain time period or a certain region.





### 4.1.1 Drought conditions in the summer of 1309

As mentioned previously, there was a relatively low level of precipitation in central and southern Jiangsu Province in the summer of 1309. The precipitation (reconstructed value) in Zhenjiang in June was approximately 102.3 mm, which is 40% lower than the modern average. In July, there were high temperatures and low precipitation in Xinghua. In August, the

precipitation (reconstructed value) in Zhenjiang was approximately 62.1 mm, which is 56% lower than the modern average.

Another piece of evidence showing that this summer was dry was the locust plague in central and southern Jiangsu Province. Dry environments provide suitable soil and climatic conditions for the growth, development, and reproduction of locusts. Thus, droughts and locust plagues often occur concomitantly in China. There are two records relating to "locusts" in *Yunshan Diary* in early August 1309. These records involve the government's calls for people and boats to catch locusts.

The locust plagues in 1309 can also be confirmed by other historical records. According to the *History of Yuan*, locust plagues occurred in Nantong, Taizhou, Yangzhou, Nanjing, and Gaoyou in Jiangsu Province in 1309. Other provinces, such as Shandong, Hebei, Henan, Anhui, and Beijing, also suffered from locust plagues. There is a record that reads "It was a dry year with scarce food" for Beijing (He, 2009; Zhang, 2004). The cross-validation of these records suggests that the summer of 1309 was relatively dry, and locust plagues occurred in many areas in North China and the middle and lower reaches of

the Yangtze River.

### 4.1.2 The severe cold winter of 1308–1309

Due to a lack of instrument-measured data in the historical period, it is impossible to accurately reconstruct the temperature. Nonetheless, it is possible to derive some indices directly related to temperature from historical records, such as the number of snowfall days(Xiao et al., 2006; Zhang and Liang, 2017), the number of snow-cover days (Xiao et al., 2006; Yan et al.,

2014), and the snow-cover depth (Zhang and Liang, 2014, 2017). Some space-related indices, such as the southern boundary of snowfall (Wang et al., 2004) and the southern boundary of river freezing (Wang et al., 2004; Zhang and Liang, 2017), are also available when there were abundant records in different locations. Through these indices, historical cold winters can be compared with the average conditions and extremely cold years in modern times (Table 5). On this basis, the extent of coldness and the climatic characteristics in the historical period can be deduced.

In January 1309, the Jiangnan Canal was frozen for at least three days. There was a relatively thick ice layer in the canal, rendering it impossible to sail boats. Based on the hydrological characteristics of rivers in the Jiangnan region, Zhang et al. (1977) deduced that the critical temperature for the river to completely freeze in this region is approximately -13 to -15 °C. In modern years when the Jiangnan Canal was frozen (1955 and 1969), the extreme daily minimum temperatures were indeed lower than -13 °C at the nearby meteorological stations. This demonstrates that the above conclusion is reliable in

general. On this basis, it is inferred that the minimum temperature in Zhenjiang might have reached below -13 °C when the river was frozen between 3 and 5 January 1309. According to the modern instrument-measured data, there were only three





years with an extreme minimum temperature below -13 °C in the Zhenjiang–Changzhou region from 1954 to 2019. The
lowest daily minimum temperature (-17 °C) in January in this region occurred in 1955. In early January 1955, the minimum
temperatures in the middle and lower reaches of the Yangtze River reached -10 to -15 °C. Except for the main stream of the

Yangtze River, all the rivers and lakes were frozen, mostly to depths of 16–35 cm. Boats were frozen in the Han River and
the Dongting Lake (Ding, 2008; Feng et al., 1985). Evidently, the climatic conditions in January 1309 were quite similar to
those in January 1955—an intense cold wave occurred in early January, causing the lakes and rivers in the lower reaches of
the Yangtze River to freeze. However, due to a lack of data, it is impossible to determine the weather conditions in January
1309 in regions other than the Zhenjiang–Changzhou region.

In February 1309, there were eight consecutive days of snowfall and 10 consecutive days of snow cover in the Zhenjiang–
Changzhou region. The roads were buried in snow, severely affecting people's commutes. The canal was also frozen. In
modern times, on average, there are approximately 3.1 snowfall days and 2.8 snow-cover days in this region in February, far
fewer than the numbers in February 1309. Clearly, the snowfall and snow-cover conditions in February 1309 were quite rare.
According to the instrument-measured records from 1954 to 2019, the lowest daily minimum temperature (-14.2 °C) in the

Zhenjiang–Changzhou region in February occurred in 1969. In 1969, as a result of the continuous strong cold waves from
late January to early February, there were six days of snowfall at Liyang station in Changzhou. The Grand Canal was frozen
from late January to early March (Zhang et al., 2011a). The Yellow River and the Bohai Sea were also frozen (Ding, 2008).

Compared to the winter of 1308-1309, there have been a few similarly cold or even colder winters in modern times.
However, it is worth noting that the extremely cold phenomena (e.g., low temperatures, snow, and freezing) in modern

winters have mostly been caused by one or several continuous strong cold waves. For example, in 1955, temperatures were
abnormally low in January but became relatively normal in February. In 1977, the minimum temperatures in January and
February were both seemingly very low. However, this is because the strongest cold wave that year occurred between late
January and early February. By contrast, in 1309, there were two extremely strong cold waves, one in January and another in
February. As a result, the Jiangnan Canal was completely frozen in January, and the number of snowfall days and snow-

cover days were both abnormally high in February. During the interval of more than 20 days between the two cold waves,
there was a notable rise in temperature. This is an exceptionally unique case.

The winter of 1308–1309 was abnormally cold. It would have also been rare even it were to occur amid the modern climatic
background. Therefore, 1308 and 1309 may have been years that marked climate change. In eastern China, the interdecadal
average temperature anomaly is positively correlated with the temperature anomaly in abnormally cold years at the

significance level of 0.001 (Zheng et al., 2005). Thus, it can be inferred that the 1300s was a relatively cold decade on an
interdecadal scale.



### 4.1.3 Transition from the MWP to the LIA in the early 14th century

The MWP approximately corresponds to the 9th–13th century in China, i.e., the period from the Five Dynasties period to the early Yuan Dynasty (Ge et al., 2013; Man, 1999). In this period, the 13th century was the warmest, comparable to the 20th
century (Ge et al., 2002b). The LIA refers to the cold period following the MWP. Most scholars agree that the LIA began at least after the 15th century in China, approximately corresponding to the Ming and Qing Dynasties. Hence, the LIA is also referred to as the "the Little Ice Age of the Ming and Qing Dynasties" (Wang, 1995; Wang et al., 2006; Zhang et al., 2013b).

*Yunshan Diary* was written in the early 14th century when the climate was turning from the MWP to the LIA. According to the winter-half-year temperature sequence for eastern China for the past two millennia, the most rapid decrease in
temperature (at a rate of 1.4 °C/90a) occurred between the mid- and late-13th century and the early 14th century. The 30 years between 1290 and 1320 were the key period when the temperature anomaly turned from positive to negative (Ge et al., 2002a; Ge et al., 2002b). The northern planting boundary of citrus recorded in *Yunshan Diary* was notably more southern than that in the mid-13th century. This demonstrates that on a multi-decadal scale, the climate in the early 14th century was no longer as warm as that at the height of the MWP. In northern China, cold disasters, such as snowstorms, have been frequent
since the 14th century (Hao et al., 2009). These pieces of evidence reflect the continuous climate-cooling process when the MWP was turning to the LIA.

### 4.2 Advantages and disadvantages of climate records in ancient diaries

Compared to other types of documentary data used in climate reconstruction, the records in private diaries have the following advantages. (1) Diaries are reliable and veracious. This is because the content of a diary is the author's own
experience, written at the time when things occurred. In addition, diary authors did not need to take into consideration any other factors (e.g., political factors) when writing their diaries and were therefore able to objectively record natural phenomena. (2) The records in diaries have little uncertainty in time and location. In *Yunshan Diary*, every record is accompanied by a clear date. The location can be deduced from the author's life experience and travel routes, which are recorded in the diary. (3) The records have a high temporal resolution and can supply daily or even sub-daily weather
information. For example, one record in *Yunshan Diary* reads, "It was sunny and warm in the daytime and rained heavily at night." (4) Diaries can reflect human interactions with climate, including the impact of climate on humans and the responses of humans to climate. For example, some records in *Yunshan Diary* show that people changed clothes when it was getting cooler. (5) Diaries are seasonally continuous and complete. By contrast, history books, local gazettes, and agricultural books are often focussed on climatic characteristics during farming seasons as well as meteorological events that affected
agriculture and social stability (Adamson, 2015; Gong et al., 1983a; Linderholm and Molin, 2005; Pfister et al., 1999; Pillatt, 2012; Zhang et al., 2007a).

Private diaries also have some disadvantages. (1) Diaries are relatively subjective. Owing to personal experience and character, authors differ in their attention and sensitivity to weather. (2) Diaries often have problems of missing records and



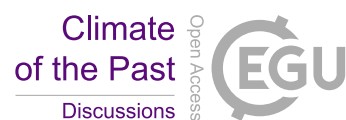

changes in the record location. These problems are notable in *Yunshan Diary*. Due to location inconsistencies, it is

impossible to reconstruct continuous climate series. Only the time periods with consistent locations are selected for analysis. (3) Diaries cover short periods of time. Limited by the life span and living conditions of the author, diary records cover several months at the shortest and several decades at the longest (Adamson, 2015; Huang et al., 2013; Pfister et al., 1999, 2008).

In summary, in historical climate research, private diaries are suitable for reconstructing short-term, high-resolution climate

series, extreme climatic events, and specific meteorological processes. They are also useful in studying the impact of climate on human lives and the responses of humans to climate. In addition, diaries can serve as supplemental data for long-term climate research.

## 5 Conclusions

Private diaries are a main type of documentary evidence for studying historical climate change. They have a number of

advantages, including high veracity and reliability, accurate time and location information, high temporal resolution, seasonal integrity, and rich content. On the other hand, private diaries have the limitations of strong subjectivity, short recording periods, missing records, and location inconsistencies. Therefore, when extracting records from diaries, it is necessary to carefully examine and assess the information. Each record should contain three clear elements, namely, time, location, and climatic/weather event. In climate reconstruction, it is critical to select suitable quantification indices and

research methods, eliminate as much interference from the author's subjectivity as possible, and highlight objective natural phenomena and their climatic significance. In particular, diaries are suitable for reconstructing short-term, high-resolution climate series, reconstructing extreme climatic events, and supplementing data for research on long-term climate change.

Climatic information in diaries mainly includes species distribution records, phenological records, weather records, and perception records. Species distribution records are adapted to average climatic conditions in the region and reflect climatic

characteristics on a multi-decadal scale. They are suitable for analysing long-term climate change. With annual- or seasonal-scale resolution, phenological records reflect interannual climatic characteristics. Daily weather records are the unique advantages of private diaries and can be used to reconstruct short-term meteorological processes or high-resolution climate series. Perception records are relatively subjective and can be used to supplement weather records. The comparison of diary records with modern instrument-measured data or other documentary data can help clarify their climatic significance.

This article presents a case study of an ancient Chinese diary of the Yuan Dynasty—*Yunshan Diary*. The records relating to weather and climate in this diary were extracted, and their climate significance was analysed. The following conclusions are drawn. (1) In the summer of 1309, the precipitation was low in central and southern Jiangsu Province. This region also suffered from a locust plague. (2) The winter of 1308–1309 was abnormally cold. In this winter, there were at least four cold waves in the TLB. In January 1309, the Jiangnan Canal was completely frozen, and the minimum temperature might have

reached -13 °C. In February 1309, there were eight consecutive days of snowfall and 10 consecutive days of snow cover in the Zhenjiang–Changzhou region, both of which are far greater than modern averages. (3) In the early 14<sup>th</sup> century at the latest, the climate in eastern China had begun to turn cold. This reflects the transition from the MWP to the LIA.

**Data availability**

All the data used to perform the analysis in this study are described and properly referenced in the paper. Yunshan Diary is
available in *A Series of Diaries of the Jin and Yuan Dynasties* published by Shanghai Bookstore Publishing House (Gu and Li, 2013). Most of the modern meteorological data are available in China Meteorological Data Service Center (http://data.cma.cn/).

**Author contributions**

SC collected data, do statistics and performed most of the analysis with guidance of YS. YS designed the research method,
supervised the study and assisted with interpreting the results. XF had the idea for the study, defined the outline of this manuscript, and made some revisions. SC and JH made the figures and drafted the manuscript. All authors participated in the analysis, provided critical feedback and helped to improve the paper.

**Competing interests**

The authors declare that they have no conflict of interest.

**Acknowledgements**

We thank all colleagues in our research group in Beijing Normal University, who provided valuable discussions and suggestions.

**Financial support**

This research has been supported by National Natural Sciences Foundation of China (No. 41771572) and National Key
Research and Development Program of China (No. 2018YFA0605602).





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





**Table 1: Phenological records in *Yunshan Diary***

| Gregorian date | Location | Phenological phenomenon | Textual descriptions |
|---|---|---|---|
| 29 November 1308 | Huzhou | Rice harvesting | *I heard my neighbours threshing rice grains. The children sang the songs of Wu without stopping throughout the night, which I did not hear in my hometown. It was a scene of a bumper harvest year.* |
| 14 December 1308 | Huzhou | First snow | *It was raining and graupelling, so I could not go out. ... After drinking, it graupelled heavily again.* |
| 21 December 1308 | Huzhou | First freezing | *There was frost. The water was frozen.* |
| 9 March 1309 | Zhenjiang | Last snow | *Snow fell heavily after the light was lit.* |
| 29 March 1309 | Zhenjiang | Full flowering of peach and plum flowers | *I looked outside of the Dingbo Gate, seeing peach and plum flowers, red and white.* |
| 10 July 1309 | Yangzhou-Gaoyou | Full flowering of lotus flowers | *Today, I travelled on a boat. There were red and white lotus flowers in the shallow water, never-ending over tens of li. This was a marvellous spectacle.* |






**Table 2: Monthly statistics of the number of days for each weather type**

| Month | Main locations | Sunny | Precipitation | Cloudy | Other | Missing | Integrity rate |
|---|---|---|---|---|---|---|---|
| September 1308* | Zhenjiang | 12 | 4 | 1 | 0 | 2 | 89.5% |
| October 1308 | Hangzhou | 8 | 16 | 3 | 1 | 3 | 90.3% |
| November 1308 | Hangzhou and Huzhou | 20 | 1 | 1 | 2 | 6 | 80.0% |
| December 1308 | Huzhou | 8 | 9 | 2 | 4 | 8 | 74.2% |
| January 1309 | Changzhou and Zhenjiang | 15 | 2 | 2 | 3 | 9 | 71.0% |
| February 1309 | Changzhou and Zhenjiang | 10 | 9 | 3 | 2 | 4 | 85.7% |
| March 1309* | Zhenjiang | 11 | 9 | 2 | 1 | 8 | 74.2% |
| April 1309 | Zhenjiang | 11 | 7 | 2 | 0 | 10 | 66.7% |
| May 1309 | Changzhou and Zhenjiang | 11 | 6 | 0 | 0 | 14 | 54.8% |
| June 1309* | Zhenjiang | 7 | 7 | 0 | 0 | 16 | 46.7% |
| July 1309 | Zhenjiang and Xinghua | 3 | 3 | 0 | 1 | 24 | 22.6% |
| August 1309 | Zhenjiang | 11 | 6 | 0 | 1 | 13 | 58.1% |
| September 1309* | Zhenjiang | 4 | 8 | 2 | 0 | 16 | 46.7% |
| October 1309 | Zhenjiang and Changzhou | 23 | 4 | 0 | 0 | 4 | 87.1% |
| November 1309* | Zhenjiang | 15 | 4 | 4 | 0 | 7 | 76.7% |
| December 1309 | Zhenjiang | 0 | 1 | 1 | 0 | 0 | 100.0% |

Notes: "Main locations" refers to the locations where Guo stayed more than three days within the month. The locations Guo passed by but did not stay at are not included. * indicates that Guo stayed at only one location for the entire month. *Yunshan Diary* covers the period from 12 September 1308 to 2 December 1309. Thus, records are available for only 19 days in September 1308 and two days in December 1309.





**Table 3: Linear regression equations between P and R for each month**

| Dependent variable | Independent variable | Regression equation | $R^2$ | $F$ | $P$ |
|---|---|---|---|---|---|
| $P_1$ | $R_1$ | $P_1 = 155.88R_1$ | 0.819 | 285.2 | 0.000 |
| $P_3$ | $R_3$ | $P_3 = 198.54R_3$ | 0.844 | 341.4 | 0.000 |
| $P_4$ | $R_4$ | $P_4 = 226.30R_4$ | 0.867 | 412.3 | 0.000 |
| $P_5$ | $R_5$ | $P_5 = 261.90R_5$ | 0.825 | 296.8 | 0.000 |
| $P_6$ | $R_6$ | $P_6 = 438.31R_6$ | 0.842 | 334.8 | 0.000 |
| $P_8$ | $R_8$ | $P_8 = 419.42R_8$ | 0.823 | 287.4 | 0.000 |
| $P_9$ | $R_9$ | $P_9 = 311.93R_9$ | 0.771 | 208.7 | 0.000 |
| $P_{10}$ | $R_{10}$ | $P_{10} = 252.92R_{10}$ | 0.789 | 232.0 | 0.000 |
| $P_{11}$ | $R_{11}$ | $P_{11} = 214.08R_{11}$ | 0.822 | 290.3 | 0.000 |

Notes: $P_i$ is the precipitation of month i (unit: mm); $R_i$ is the proportion of precipitation days of month i; $R^2$, $F$, and $P$ are the goodness-of-fit, the ratio of the regression mean square to the residual means square ratio, and the significance level of the regression equation, respectively.


**Table 4: Daily weather conditions in the period December 1308–February 1309**

| Region | | | | | | | | | | |
|---|---|---|---|---|---|---|---|---|---|---|
| **Huzhou region** | **12.1** | **12.2** | **12.3** | **12.4** | **12.5** | 12.6 | 12.7 | 12.8 | 12.9 | 12.10 |
| | Rain | Rain | Sunny | Frost | Frost | Missing | Sunny | Sunny | Sunny | Sunny |
| | 12.11 | 12.12 | 12.13 | 12.14 | 12.15 | 12.16 | 12.17 | 12.18 | 12.19 | 12.20 |
| | Rain | Rain | Missing | Snow | Snow | Rain | Rain | Snow | Overcast | Sunny |
| | **12.21** | 12.22 | 12.23 | 12.24 | 12.25 | 12.26 | 12.27 | 12.28 | 12.29 | 12.30 |
| | Freeze | Missing | Missing | Frost | Sunny | Overcast | Missing | Missing | Missing | Wind |
| **Zhenjiang–Changzhou region** | 12.31 | **1.1** | **1.2** | **1.3** | **1.4** | **1.5** | 1.6 | 1.7 | 1.8 | 1.9 |
| | Missing | Snow | Freeze | Freeze | Freeze | Freeze | Sunny | Sunny | Sunny | Sunny |
| | 1.10 | 1.11 | 1.12 | 1.13 | 1.14 | 1.15 | 1.16 | 1.17 | 1.18 | 1.19 |
| | Sunny | Sunny | Sunny | Sunny | Missing | Sunny | Overcast | Sunny | Sunny | Missing |
| | 1.20 | 1.21 | 1.22 | 1.23 | 1.24 | 1.25 | 1.26 | 1.27 | 1.28 | 1.29 |
| | Sunny | Sunny | Missing | Missing | Missing | Missing | Missing | Wind | Missing | Missing |
| | 1.30 | 1.31 | **2.1** | **2.2** | **2.3** | **2.4** | **2.5** | **2.6** | **2.7** | **2.8** |
| | Overcast | Rain | Snow | Snow | Snow | Snow | Snow | Snow | Snow | Snow |
| | **2.9** | **2.10** | 2.11 | 2.12 | 2.13 | 2.14 | 2.15 | 2.16 | 2.17 | 2.18 |
| | Freeze | Freeze | Missing | Sunny | Rain | Overcast | Missing | Missing | Sunny | Overcast |
| | 2.19 | 2.20 | 2.21 | 2.22 | 2.23 | 2.24 | 2.25 | 2.26 | 2.27 | 2.28 |
| | Overcast | Sunny | Sunny | Sunny | Sunny | Sunny | Missing | Sunny | Sunny | Sunny |

| Legend | Sunny | Overcast | Rain | Snow | Frost | Wind | Freeze | Missing | Cold wave |
|---|---|---|---|---|---|---|---|---|---|

Notes: "Freeze" means the freezing of water. In fact, "frost" and "freeze" are natural phenomena affected by weather, which are closely related to temperature. Therefore, they are also shown in this table.





**Table 5: Climatic indices of the Zhenjiang–Changzhou region in January and February 1309 and their comparison with those obtained from modern instrument-measured records**

|  | 1309 | Average for 1953–2010 | Modern extreme cold-winter years | | |
|---|---|---|---|---|---|
|  |  |  | 1955 | 1969 | 1977 |
| Ns in January (days) | 1 | 3.8 | 5 | 5.5 | 14.5 |
| Rs in January | 33.33% | 38.10% | 71.42% | 42.31% | 93.55% |
| Ns in February (days) | 8 | 3.1 | 5 | 9.5 | 2.5 |
| Rs in February | 88.89% | 28.77% | 33.33% | 59.38% | 38.46% |
| Minimum temperature in January (°C) | <-13 | -6.7 | -17 | -8.4 | -15.3 |
| Minimum temperature in February (°C) | - | -5.3 | -4.9 | -14.2 | -12.2 |
| Average temperature in January (°C) | - | 2.8 | -0.7 | 2.1 | -0.7 |
| Average temperature in February (°C) | - | 4.6 | 6.2 | 0.7 | 2.4 |

Note: Ns = number of snowfall days, and Np = number of precipitation days (including rainfall days and snowfall days). Rs = rate of snowfall = $\frac{Ns}{Np}$ (Gong et al., 1983b).






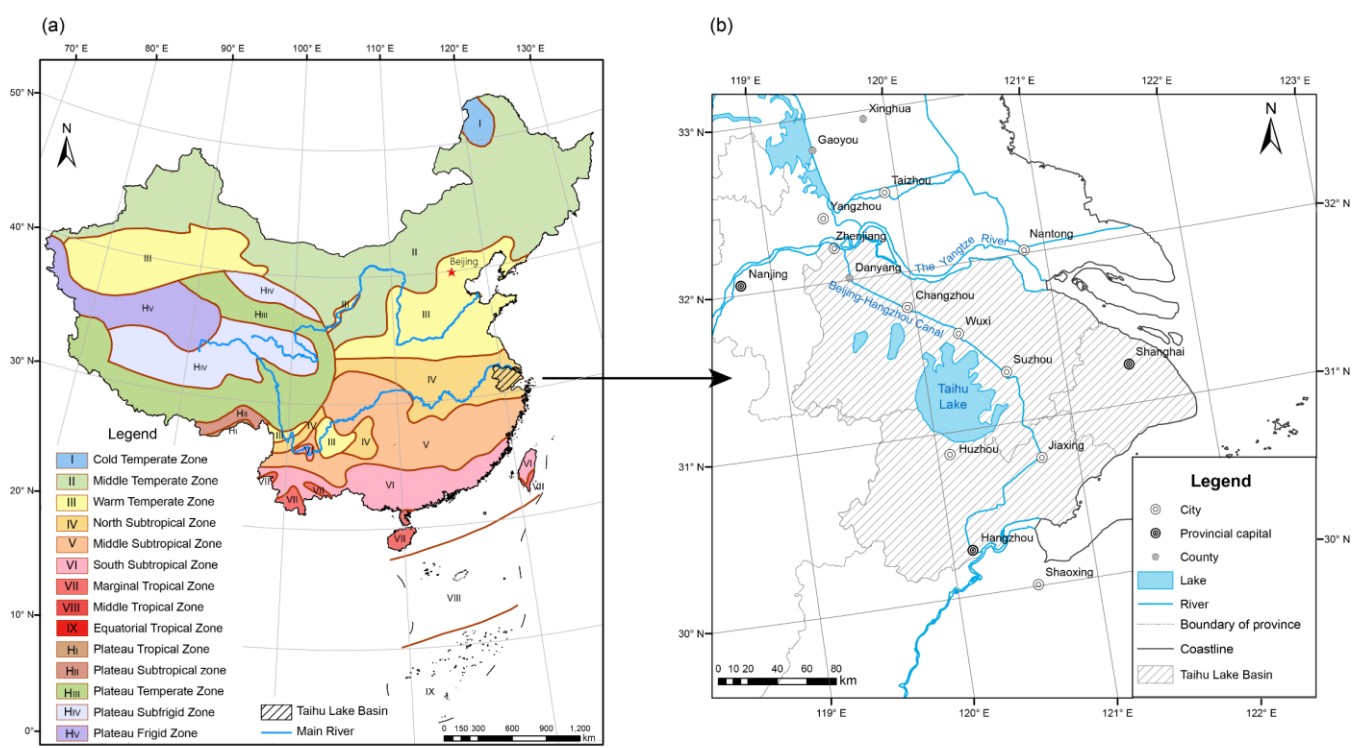

**Figure 1: Map of the study area. (a) Climatic regionalization of China and location of the TLB (Wang and Zuo, 2010). (b) The scope of Guo's travelling. (c) Guo's travelling route.**







**Figure 2: Precipitation and the proportion of precipitation days in Zhenjiang for each month of 1309 and their comparison with the modern averages**

Notes: * indicates a month when Guo stayed entirely in Zhenjiang. Guo stayed in Zhenjiang fewer than 20 days in February, July, and December. Therefore, the precipitation was not reconstructed for these three months.