# Peer review of "Climate records in ancient Chinese diaries and their application in historical climate reconstruction—A case study of *Yunshan Diary"

_Climate of the Past, 2020_

## Referee Comment (RC1) · Anonymous Referee #1 · 8 Jun 2020

This is a detailed analysis that extracts a lot of information from a relatively short diary. I certainly recommend its publication. I have recommended major revisions only due to the comment below, which will require a small amount of new analysis:

The justification for selection an intercept of 0 for the regression equations is undermined by the observation that rainfall can occur even when it is not observed by humans. I suggest re-calculating the regressions. I don't imagine this will change the conclusions or affect the analysis much, although figure 2 will need to be redrawn.

Otherwise my points are relatively minor:

Line 33 – please define 'ancient diaries'

[Figure]

Line 56-58 – give a summary here of the key findings.

Lines 89-91 – it's not clear to me what this delineation of four categories is adding, given that the information from the diary are then delineated into four different. Please either clarify or consider removing this sentence.

Lines 128-129 and 368-372 – you need to define what you mean by the MWP (and LIA) before you use this term for the first time. Please move the section in lines 368-372 into the opening sections of the paper.

Section 3.2 – This section is misleadingly titles. Phenology is the seasonal timing of biological phenomena, whereas this section also looks at first and last snows. A more accurate title would be something like 'climate-dependent phenomena' or 'documentary proxy data', both of which would include phenological records.

Lines 229-230 – Please give the justification for this threshold of 20 days

Table 3 – If you are going to express P values to 4 significant figures, please express 0.000 as >0.001 – P = 0 is a misnomer. Please explain why February, July and December are missing here (i.e. before Figure 2)

Lines 255-259 – this is a good point, but it rather undermines the decision to put the intercept of the regression equations through zero, since R = 0 does not necessarily equate to P = 0. I suggest you redo your regression equations. Clearly if you're getting ridiculous intercepts (e.g. P = 100 mm) then you may be justified to force the intercept through zero, but you then need to be clear that this is what you are doing.

Line 361 – Remove the sentence 'This is an exceptionally unique case'. You don't really have the evidence to say that, it could have happened many times between 1309 and 1955.

Lines 362-366 – I think you have to be very careful in assuming climatic changes when you only have the weather for a couple of years. Please give detail of the analysis and findings in Zheng et al. (2005), otherwise it's difficult to say whether your inference is

justified.

Lines 409-424 – this mostly repeats what was in the opening sections, and can either be significantly shortened or removed altogether.

---

## Referee Comment (RC2) · Anonymous Referee #2 · 11 Jun 2020

The paper focused on how to apply historical documents properly in the reconstruction of climate. The author selected Yunshan Diary, including species distribution records, phenological records, evaluated the advantage for climate reconstruction, meanwhile, they noted the shortcomings of personal diaries, e.g., limitations of strong subjectivity, short recording periods, missing records, and location inconsistencies. Those analyses are valuable for the other paleo-scholars when they use the historical archives to study the climate change. After that, they reconstructed the severe cold winter of 1308–1309 and drought of 1309. The result is meaningful for understanding the transition from the Medieval Warm Period to the Little Ice Age in the Yangtze River Basin in China. Actually, the dairy was written in 1290-1320, which is important period when climate was

changing from warm to cold. Thus, finding more valuable related climate information from Yunshan diary is expected in the future. Before the acceptance for publication, minor revisions should be done: Major comments: 1. L 63, more descriptions on Guo Bi are necessary to added in this part, for example, his official position, life track etc, which will convince the readers in the reconstructions and records. 2. L93 Four types of related information "weather records, meteorological disaster records, phenological records, and records relating to the characteristics of regional climate", please provide some example to tell readers what they were. And your full extraction method (including recognizing the missing or wrong records) needs to be given. 3. L230-L235: Figure2 about precipitation and total number of rain days, in order to understanding the characteristics of precipitation in modern period, I would suggest to plot February, July and December information, and combine upper and bottom figure together, which is helpful to see the relationship between rain days and precipitation clearly. And how about other years, is there any month with plenty of rain from 1290 to 1320? Could you provide a case? 4. L363, "1308 and 1309 may have been years that marked climate change". This infer is arbitrary decision, it needs more evidences to support.

Minor comments: (1) L73, The annual average precipitation here is 1,000–1,400 mm, the annual average temperature is 15–16 °C, and the average temperature in January 1.5–4 °C. Please add the reference period, it is important to judge the cold/warm or dry/wet condition comparing a certain period (2) L75 All the dates in this article are Gregorian dates. Here please show an example to readers how the date was converted. (3)In Section 4.1.1, the authors mentioned "the modern average" precipitation for several times. To be specific, which time period does it refer to? (4) In Table 2, there is a column named "Other". What is the meaning? Does it represent some specific types of weather? (5)Table 3, the number of sample size and stated period are need to be provided.

---

## Author Comment (AC1) · 1 Jul 2020

Dear Editors and Referees,

Thank you very much for taking your time to review this manuscript. We really appreciate all your comments and suggestions. They have enabled us to improve our work. According to the comments, we did some modifications. Please find our itemized responses below. Thanks again!

Comment 1: The justification for selection an intercept of 0 for the regression equations is undermined by the observation that rainfall can occur even when it is not observed

by humans. I suggest re-calculating the regressions. I don't imagine this will change the conclusions or affect the analysis much, although figure 2 will need to be redrawn.

Response 1: Thank you for the comment. We rechecked the precipitation reconstruction process. The data used to establish the regression equations are not from human observation. Instead, the data of precipitation and precipitation days are both modern meteorological data from instrumental measurements. They come from the dataset of monthly climate data of Chinese surface stations (dataset code: SURF_CLI_CHN_MUL_MON), which is published by China Meteorological Data Service Center (http://data.cma.cn/). The 'precipitation days' is defined as days with daily precipitation $\geq$0.1mm. So we think the point that rainfall can occur even when it is not observed by humans does not conflict with the selection of regression equations with an intercept of 0.

Comment 2: Line 33 – please define 'ancient diaries'

Response 2: According to the comment, the definition of 'ancient diaries' was added: "Ancient diaries refers to diaries written before the collapse of the Qing Dynasty in 1912".

Comment 3:Line 56-58 – give a summary here of the key findings.

Response 3: According to the comment, the key findings of the case study of Yunshan Diary were briefly added: "Through the analysis of Yunshan Diary, the severe cold winter of 1308/1309 in the Taihu Lake Basin and the drought in the summer of 1309 in southern Jiangsu Province are identified. On a multi-decadal scale, it is proved that the climate had begun to turn cold in the early 14th century at the latest."

Comment 4: Lines 89-91 – it's not clear to me what this delineation of four categories is adding, given that the information from the diary are then delineated into four different. Please either clarify or consider removing this sentence.

Response 4: Thank you for the comment. It is a general method to divide climate

information into the four categories mentioned here, which is applicable to most Chinese historical documents. For better understanding, we added some explanations and examples of the four categories in the manuscript. The revised sentence is as follows: "On the whole, climate records in historical documents can be classified into four categories based on the content, (1) weather records, including qualitative descriptions (such as sunny, cloudy, rainy) and quantitative observation (such as the infiltration depth of each precipitation event in Yu Xue Fen Cun); (2) meteorological disaster records, such as floods, droughts and their impacts on agriculture and society; (3) phenological records, such as the flowering date of plants, the migration date of birds; (4) records relating to the characteristics of regional climate, such as cropping system, distribution range of specific crops and fruits, the southern boundary of snowfall and the southern boundary of river freezing." As for the four types of climate information in Yunshan Diary, species distribution records, phenological records and weather records belong to the fourth category, the third category and the first category mentioned in the previous paragraph respectively. The perception record is a particular type only appearing in diaries, and we added the corresponding explanation in the manuscript: "As one kind of private documents, the diary contains perception records additionally, which record authors' subjective feelings about the weather (such as warm and cold)."

Comment 5: Lines 128-129 and 368-372 – you need to define what you mean by the MWP (and LIA) before you use this term for the first time. Please move the section in lines 368-372 into the opening sections of the paper.

Response 5: Thank you for the comment. We considered how to solve the problem. Lines 368-372 are literature reviews of the starting and ending time of MWP and LIA in China, which is closely related to the discussion of climate transition time in the following paragraphs. If they were moved into the opening sections, the integrity of section 4.1.3 would be undermined. In order to avoid the problem of unclear definitions of MWP and LIA, the previous references to the two terms were deleted. All issues related to MWP and LIA were discussed in section 4.1.3.

Comment 6: Section 3.2 – This section is misleadingly titles. Phenology is the seasonal timing of biological phenomena, whereas this section also looks at first and last snows. A more accurate title would be something like 'climate-dependent phenomena' or 'documentary proxy data', both of which would include phenological records.

Response 6: We are grateful for the comment. This problem may be caused by the different definitions of "phenology". The original definition of phenology was the study of the timing of recurring biological events (Leith, 1974). But in recent years, some scholars have proposed that phenology is the study of the times of recurring natural phenomena (van Vliet and De Groot, 2003). It includes not only biotic phenology (observations of plant and animals), but also abiotic phenology (observations of other natural phenomena with seasonal character) (Jeanneret and Rutishauser, 2010). In this paper, the broad definition is adopted. For better understanding, a clear explanation of the meaning of "Phenology" was added: "Here, the broad definition is adopted, that is, phenological phenomena not only includes recurrent biological phenomena, but also recurrent meteorological or hydrological phenomena, such as the timing of frost, snow and river freezing."

Comment 7: Lines 229-230 – Please give the justification for this threshold of 20 days

Response 7: Thank you for pointing out this deficiency. There is indeed not enough justification to use 20 days as a threshold for participation in precipitation reconstruction. Therefore, the original sentence was deleted and the reasons for excluding February, July and December were explained separately.

Comment 8: Table 3 – If you are going to express P values to 4 significant figures, please express 0.000 as >0.001 – P = 0 is a misnomer. Please explain why February, July and December are missing here (i.e. before Figure 2)

Response 8: Thank you for pointing out this deficiency. In the 'P' column of Table 3, 0.000 was changed to <0.001. The reason why February, July and December are missing was explained in the main text.

Comment 9: Lines 255-259 – this is a good point, but it rather undermines the decision to put the intercept of the regression equations through zero, since R = 0 does not necessarily equate to P = 0. I suggest you redo your regression equations. Clearly if you're getting ridiculous intercepts (e.g. P = 100 mm) then you may be justified to force the intercept through zero, but you then need to be clear that this is what you are doing.

Response 9: Thanks for the comment and we want to do some explanation. The perspective stated here is some precipitation events might have been too slight for humans to notice, but the data of precipitation and precipitation days used to establish the regression equations is modern meteorological data from instrumental measurements. Therefore, we think that this point does not conflict with the selection of the regression equation with an intercept of 0. In addition, a new perspective was added here: "precipitation events occurred in the night may also be ignored by humans".

Comment 10: Line 361 – Remove the sentence 'This is an exceptionally unique case'. You don't really have the evidence to say that, it could have happened many times between 1309 and 1955.

Response 10: Thanks for the comment and we did some modifications accordingly. The original expression was indeed too absolute. But we still wanted to clarify that the winter of 1308/1309 is different compared to modern cold winters, so the sentence was changed to "it is quite unusal".

Comment 11: Lines 362-366 – I think you have to be very careful in assuming climatic changes when you only have the weather for a couple of years. Please give detail of the analysis and findings in Zheng et al. (2005), otherwise it's difficult to say whether your inference is justified.

Response 11: According to the comment, a detailed explanation of the conclusion in Zheng et al. (2005) was added: "Zheng et al. (2005) analysed the winter temperature anomaly sequence in eastern China from 1951 to 1995. The results show that the average temperature anomaly of every 10 years is positively correlated with the

temperature anomalies of abnormally cold years among the 10 years. The correlation coefficient is 0.965 and the significance level is 0.001."

Comment 12: Lines 409-424 – this mostly repeats what was in the opening sections, and can either be significantly shortened or removed altogether.

Response 12: Thanks for the suggestion. Accordingly, this part was simplified and those repetitive sentences were deleted.

Please also note the supplement to this comment:
https://cp.copernicus.org/preprints/cp-2020-72/cp-2020-72-AC1-supplement.pdf

**Supplement:**

**References about the phenology (corresponding to comment 6)**

Jeanneret François and Rutishauser This, Seasonality as a Core Business of Phenology, in: Phenological Research: Methods for Environmental and Climate Change Analysis, edited by Hudson IL and Keatley MR, Springer, Dordrech, Netherlands, 2010.

Leith H: Phenology and seasonal modeling, Springer-Verlag, New York, USA, 1974.

van Vliet AJH and De Groot RS: "Challenging times" in the context of the European phenology network, in: Challenging times: towards an operational system for monitoring, modeling, and forecasting of phenological changes and their socio-economic impacts, edited by van Vliet AJH, Wageningen University, Wageningen, Netherlands, 2003.

---

## Author Comment (AC2) · 1 Jul 2020

Dear Editors and Referees,

Thank you very much for taking your time to review this manuscript. We really appreciate all your comments and suggestions. They have enabled us to improve our work. According to the comments, we did some modifications. Please find our itemized responses below.

Thanks again!

Comment 1: L63, more descriptions on Guo Bi are necessary to added in this part,

for example, his official position, life track etc, which will convince the readers in the reconstructions and records.

Response 1: According to the comment, a brief introduction to Guo's official position and life track was added: "Guo was born in present-day Zhenjiang in Jiangsu Province. He used to be a lecturer at local academies in Jiangsu and Jiangxi, and a county official in Zhejiang. During the period recorded in Yunshan Diary, Guo resided in Zhenjiang, but he travelled multiple times on official business or to visit friends.".

Comment 2: L93 Four types of related information "weather records, meteorological disaster records, phenological records, and records relating to the characteristics of regional climate", please provide some example to tell readers what they were. And your full extraction method (including recognizing the missing or wrong records) needs to be given.

Response 2: Thank you for the comment. The explanations and examples of the four categories were added. The revised sentence is as follows: "On the whole, climate records in historical documents can be classified into four categories based on the content, (1) weather records, including qualitative descriptions (such as sunny, cloudy, rainy) and quantitative observation (such as the infiltration depth of each precipitation event in Yu Xue Fen Cun); (2) meteorological disaster records, such as floods, droughts and their impacts on agriculture and society; (3) phenological records, such as the flowering date of plants, the migration date of birds; (4) records relating to the characteristics of regional climate, such as cropping system, distribution range of specific crops and fruits, the southern boundary of snowfall and the southern boundary of river freezing." The extraction methods of various types of records were described in section 3.1.1, 3.2.1, 3.3.1 and 3.3.2 separately. The method for recognizing missing records was added in section 3.3.1.

Comment 3: L230-L235: Figure2 about precipitation and total number of rain days, in order to understanding the characteristics of precipitation in modern period, I would

suggest to plot February, July and December information, and combine upper and bottom figure together, which is helpful to see the relationship between rain days and precipitation clearly. And how about other years, is there any month with plenty of rain from 1290 to 1320? Could you provide a case?

Response 3: According to the suggestion, modern precipitation information for February, July and December was added in Figure 2. And the upper and bottom figure was combined together. The modified figure is shown below as Fig.1. However, due to the lack of proxy data, there is no available case study of precipitation reconstruction in 1290 to 1320.

Comment 4: L363, "1308 and 1309 may have been years that marked climate change". This infer is arbitrary decision, it needs more evidences to support.

Response 4: Thank you for pointing out this deficiency. The inference is indeed too absolute and lacks sufficient evidence to support it. So the sentence was deleted.

Comment 5: L73, The annual average precipitation here is 1,000–1,400 mm, the annual average temperature is 15–16°C, and the average temperature in January 1.5–4°C. Please add the reference period, it is important to judge the cold/warm or dry/wet condition comparing a certain period.

Response 5: Thank you for the comment. The reference period is 1951-1980, and we added it in the manuscript.

Comment 6: L75 All the dates in this article are Gregorian dates. Here please show an example to readers how the date was converted.

Response 6: According to the suggestion, an example of date conversion was added: "For example, a record reads 'It is sunny on the second day of the ninth lunar month of the first year of Zhida', and the date can be converted to 16 September 1308."

Comment 7: In Section 4.1.1, the authors mentioned "the modern average" precipitation for several times. To be specific, which time period does it refer to?

Response 7: Thanks for the comment. The reference period is 1981-2010 and we added it in the manuscript.

Comment 8: In Table 2, there is a column named "Other". What is the meaning? Does it represent some specific types of weather?

Response 8: Thanks for the comment. "Other" means weather records except for sunny, cloudy and precipitation, such as windy, frost, fog and so on. The explanation was added in the manuscript.

Comment 9: Table 3, the number of sample size and stated period are need to be provided.

Response 9: We are grateful for the comment. The data source, sample size and stated period of table 3 were all stated in the main text (Lines 236-239). So we didn't state it again in the footnote of table 3.
* * *
**P (1309)**    **P (Modern average)**    **R (1309)**    **R (Modern average)**

**Fig. 1.** Precipitation and the proportion of precipitation days in Zhenjiang for each month of 1309 and their comparison with the modern averages

---

## Author Response (AR1)

**Response to the comments on"Climate records in ancient Chinese diaries and their application in historical climate reconstruction—A case study of *Yunshan Diary*"**

Dear Editors and Reviewers,

Thank you very much for taking your time to review this manuscript. We really appreciate all your comments and suggestions. They have enabled us to improve our work. According to the comments, we did some modifications. Please find our itemized responses below. The revised manuscript is attached behind, and the main modifications were marked in red.

Thanks again!

**1. Comments from editors**

Comment 1: on page 3, line 68: It is not clear what is meant by "encompasses an area of 3.6 x 104 km2." I would expect area to be measured either in two dimensions (i.e., Y x Z km) or in km2. I also thought that Lake Tai was itself about 2000 km2, so the basin would be larger.

Response: Thank you for pointing out this deficiency. It was a clerical error in the previous manuscript, and it should be "encompasses an area of 3.69 x $10^4$ km$^2$." We have corrected it (Line 72).

Comment 2: on page 4, line 94 and in sections 3.3.1 and 3.3.2: Please find clearer and more distinct terms for "weather records" and "perception records". Based on the explanations in the text, I would suggest "daily weather descriptions" and "personal experiences of meteorological conditions".

Response: According to the comment, "weather records" was changed to "daily weather descriptions", and "perception records" was changed to "personal experiences of meteorological conditions" in abstract, section 3 and section 5.

Comment 3: The citation on page 18, line 547 should be: "Pfister, C., Brázdil, R., Glaser, R., Bokwa, A., ...." That is, the third author's family name is Glaser and his first initial is R.

Response: Thank you for pointing out the mistake. The citation was changed to "Pfister, C., Brázdil, R., Glaser, R., Bokwa, A., …" (Lines 557-559).

Comment 4: p22: "graupel" is a very rare word in English, but it is used appropriately here. The journal editor may mark this word as a mistake, but you may keep it.

Response: We are grateful for the comment. For the accuracy of the text, we would like to keep the word "graupel".

**2. Comments from referee#1**

Comment 1: The justification for selection an intercept of 0 for the regression equations is undermined by the observation that rainfall can occur even when it is not observed by humans. I suggest re-calculating the regressions. I don't imagine this will change the conclusions or affect the analysis much, although figure 2 will need to be redrawn.

Response: Thank you for the comment. We rechecked the precipitation reconstruction process. The data used to establish the regression equations are not from human observation. Instead, the data of precipitation and precipitation days are both modern meteorological data from instrumental measurements. They come from the dataset of monthly climate data from Chinese surface stations (dataset code: SURF_CLI_CHN_MUL_MON), which is published by China Meteorological Data Service Center (http://data.cma.cn/). The 'precipitation days' is defined as days with daily precipitation ≥0.1mm. So we think the point that rainfall can occur even when it is not observed by humans does not conflict with the selection of regression equations with an intercept of 0.

Comment 2: Line 33 – please define 'ancient diaries'

Response: According to the comment, the definition of 'ancient diaries' was added in Line 33.

Comment 3:Line 56-58 – give a summary here of the key findings.

Response: According to the comment, the key findings of the case study of Yunshan Diary was briefly added in Lines 58-61.

Comment 4: Lines 89-91 – it's not clear to me what this delineation of four categories is adding, given that the information from the diary are then delineated into four different. Please either clarify or consider removing this sentence.

Response: Thanks for the suggestion and we did some modifications. The delineations and examples of four categories of climate records were added in Lines 94-99. As for the four types of climate information in Yunshan Diary, species distribution records, phenological records and daily weather descriptions belong to the fourth category, the third category and the first category mentioned in the previous paragraph respectively. The personal experience of meteorological conditions is a particular type only appearing in diaries, which was explained in Lines 99-101.

Comment 5: Lines 128-129 and 368-372 – you need to define what you mean by the MWP (and LIA) before you use this term for the first time. Please move the section in lines 368-372 into the opening sections of the paper.

Response: Thank you for the comment. We considered how to solve the problem. Lines 383-388 (i.e. the original lines 368-372) are literature reviews of the starting and ending time of MWP and LIA in China, which is closely related to the discussion of climate transition time in the following paragraphs. If they were moved into the opening sections, the integrity of section 4.1.3 would be undermined. In order to avoid the problem of unclear definitions of

MWP and LIA, the previous references to the two terms were deleted. All issues related to MWP and LIA were discussed in section 4.1.3.

Comment 6: Section 3.2 – This section is misleadingly titles. Phenology is the seasonal timing of biological phenomena, whereas this section also looks at first and last snows. A more accurate title would be something like 'climate-dependent phenomena' or 'documentary proxy data', both of which would include phenological records.

Response: We are grateful for the comment. This problem may be caused by the different definitions of 'phenology'. The original definition of 'phenology' was the study of the timing of recurring biological events[1]. But in recent years, some scholars have proposed that phenology is the study of the times of recurring natural phenomena[2]. It includes not only biotic phenology (observations of plant and animals), but also abiotic phenology (observations of other natural phenomena with seasonal character)[3]. In this paper, the broad definition is adopted. For better understanding, a clear explanation of the meaning of 'Phenology' was added in Lines 151-153.

Comment 7: Lines 229-230 – Please give the justification for this threshold of 20 days

Response: Thank you for pointing out this deficiency. There is indeed not enough justification to use 20 days as a threshold for participation in precipitation reconstruction. Therefore, the original sentence was deleted and the reasons for excluding February, July and December were explained separately (Lines 242-244).

Comment 8: Table 3 – If you are going to express P values to 4 significant figures, please express 0.000 as >0.001 – P = 0 is a misnomer. Please explain why February, July and December are missing here (i.e. before Figure 2)

Response: Thank you for pointing out this deficiency. In the 'P' column of Table 3, 0.000 was changed to <0.001. The reason why February, July and December are missing was explained in Lines 242-244 in the main text.

Comment 9: Lines 255-259 – this is a good point, but it rather undermines the decision to put the intercept of the regression equations through zero, since R = 0 does not necessarily equate to P = 0. I suggest you redo your regression equations. Clearly if you're getting ridiculous intercepts (e.g. P = 100 mm) then you may be justified to force the intercept through zero, but you then need to be clear that this is what you are doing.

Response: Thanks for the comment and we want to do some explanation. The perspective stated
* * *
[1] Leith H: Phenology and seasonal modeling, Springer-Verlag, New York, USA, 1974.
[2] van Vliet AJH and De Groot RS: "Challenging times" in the context of the European phenology network, in: Challenging times: towards an operational system for monitoring, modeling, and forecasting of phenological changes and their socio-economic impacts, edited by van Vliet AJH, Wageningen University, Wageningen, Netherlands, 2003.
[3] Jeanneret François and Rutishauser This, Seasonality as a Core Business of Phenology, in: Phenological Research: Methods for Environmental and Climate Change Analysis, edited by Hudson IL and Keatley MR, Springer, Dordrech, Netherlands, 2010.

here is some precipitation events might have been too slight for humans to notice, but the data of precipitation and precipitation days used to establish the regression equations is modern meteorological data from instrumental measurements. Therefore, we think that this point does not conflict with the selection of the regression equation with an intercept of 0. In addition, a new perspective was added here: precipitation events occurred in the night may also be ignored by humans (Line 271).

Comment 10: Line 361 – Remove the sentence 'This is an exceptionally unique case'. You don't really have the evidence to say that, it could have happened many times between 1309 and 1955.

Response: Thanks for the comment and we did some modifications accordingly. The original expression was indeed too absolute. But we still wanted to clarify that the winter of 1308/1309 is different compared to modern cold winters, so the sentence was changed to 'It is quite unusal' (Line 376).

Comment 11: Lines 362-366 – I think you have to be very careful in assuming climatic changes when you only have the weather for a couple of years. Please give detail of the analysis and findings in Zheng et al. (2005), otherwise it's difficult to say whether your inference is justified.

Response: According to the comment, a detailed explanation of the conclusion in Zheng et al. (2005) was added in Lines 378-380.

Comment 12: Lines 409-424 – this mostly repeats what was in the opening sections, and can either be significantly shortened or removed altogether.

Response: Thanks for the suggestion. Accordingly, this part was simplified.

**3. Comments from referee#2**

Comment 1: L63, more descriptions on Guo Bi are necessary to added in this part, for example, his official position, life track etc, which will convince the readers in the reconstructions and records.

Response: According to the comment, a brief introduction to Guo's official position and life track was added in Lines 66-69.

Comment 2: L93 Four types of related information "weather records, meteorological disaster records, phenological records, and records relating to the characteristics of regional climate", please provide some example to tell readers what they were. And your full extraction method (including recognizing the missing or wrong records) needs to be given.

Response: Thank you for the comment. The delineations and examples of four categories of climate records were added in Lines 94-99. The extraction methods of various types of records were described in section 3.1.1, 3.2.1, 3.3.1 and 3.3.2 separately. The method for recognizing missing records was added in Line 222.

Comment 3: L230-L235: Figure2 about precipitation and total number of rain days, in order to understanding the characteristics of precipitation in modern period, I would suggest to plot February, July and December information, and combine upper and bottom figure together, which is helpful to see the relationship between rain days and precipitation clearly. And how about other years, is there any month with plenty of rain from 1290 to 1320? Could you provide a case?

Response: According to the suggestion, modern precipitation information for February, July and December was added in Figure 2. And the upper and bottom figure was combined together. However, due to the lack of proxy data, there is no available case study of precipitation reconstruction in 1290 to 1320.

Comment 4: L363, "1308 and 1309 may have been years that marked climate change". This infer is arbitrary decision, it needs more evidences to support.

Response: Thank you for pointing out this deficiency. The inference is indeed too absolute and lacks sufficient evidence to support it. So the sentence was deleted.

Comment 5: L73, The annual average precipitation here is 1,000–1,400 mm, the annual average temperature is 15–16℃, and the average temperature in January 1.5–4℃. Please add the reference period, it is important to judge the cold/warm or dry/wet condition comparing a certain period.

Response: According to the comment, the reference period was added in Line 76.

Comment 6: L75 All the dates in this article are Gregorian dates. Here please show an

example to readers how the date was converted.

Response: According to the suggestion, an example of date conversion was added in Lines 81-82.

Comment 7: In Section 4.1.1, the authors mentioned "the modern average" precipitation for several times. To be specific, which time period does it refer to?

Response: Thanks for the comment. The time period which 'the modern average' refers to was clarified in Line 257, 258, 323, and 325.

Comment 8: In Table 2, there is a column named "Other". What is the meaning? Does it represent some specific types of weather?

Response: According to the comment, the meaning of 'other' was supplemented in Lines 221-222.

Comment 9: Table 3, the number of sample size and stated period are need to be provided.

Response: We are grateful for the comment. The data source, sample size and stated period of table 3 were all stated in the main text (Lines 251-253). So we didn't state it again in the footnote of table 3.

[revised manuscript text omitted]